# Effects of *Bacillus subtilis* on Production Performance, Bone Physiological Property, and Hematology Indexes in Laying Hens

**DOI:** 10.3390/ani11072041

**Published:** 2021-07-08

**Authors:** Xinyu Zou, Sha Jiang, Mi Zhang, Haiqiang Hu, Xiaoling Wu, Jianzhu Liu, Meilan Jin, Hengwei Cheng

**Affiliations:** 1Joint International Research Laboratory of Animal Health and Animal Food Safety, College of Veterinary Medicine, Southwest University, Chongqing 400715, China; yu112055@163.com (X.Z.); zhangmi202105@163.com (M.Z.); bbb971198027@email.swu.edu.cn (X.W.); meilan0622@swu.edu.cn (M.J.); 2Immunology Research Center, Medical Research Institute, Southwest University, Chongqing 402460, China; 3College of Animal Science and Technology, Southwest University, Chongqing 400715, China; hhq970818@163.com; 4China College of Veterinary Medicine, Shandong Agricultural University, Tai’an 271018, China; liujz@sdau.edu.cn; 5Livestock Behavior Research Unit, USDA-Agricultural Research Service, West Lafayette, IN 47907, USA; cheng5@purdue.edu

**Keywords:** *Bacillus subtilis*, bone quality, eggshell, calcium, inflammation factor

## Abstract

**Simple Summary:**

Due to breeding for high egg production, laying hens are at great risk for developing osteoporosis. To develop an effective feed additive for reducing the bone damage and associated pain and economic loss has become a critical issue affecting the poultry industry. The aim of this study was to investigate the effects of *Bacillus subtills* as a feed supplement on production performance and bone pathophysiological characteristics of laying hens. The results showed that *Bacillus subtilis* increases marketable eggs, protects bone health, changes the distribution of phosphorus between blood and bone, and increases estrogen but decreases interleukin-1 and tumor necrosis factor-α concentrations in blood. Results indicate that *Bacillus subtilis* can be used as a dietary supplement to increase marketable egg production and bone health of laying hens by inhibiting gut and systemic inflammation via the microbiota-gut-immune and the microbiota-gut-bone axes.

**Abstract:**

This study was to investigate the effects of *Bacillus subtilis* on production performance and bone pathophysiological characteristics of layers. Twenty-four 48-week-old Lohmann Pink-shell laying hens were randomly divided into two groups: a basic diet (control) and the basic diet mixed with *Bacillus subtilis* (0.5 g/kg) for a 60-day trial. Statistically, independent-sample *t*-test was used to assess the treatment differences. The results showed that *Bacillus subtilis* supplementation improved the percent of marketable eggs (*p* < 0.05) with reduced numbers of broken and soft-shelled eggs but had no effects on egg weight, height of albumen, yolk color, and Haugh unit (*p* > 0.05). *Bacillus subtilis* supplement also elevated maximum load (*p* = 0.06), maximum stress (*p* = 0.01), stiffness (*p* < 0.01), and Young’s modulus (*p* < 0.01) but suppressed maximum strain (*p* = 0.06) in the femur. In addition, compared with control birds, phosphorous concentration (*p* < 0.01) was reduced in serum at day 61 but increased in the femur (*p* < 0.05) in *Bacillus subtilis* fed birds. *Bacillus subtilis* fed birds also had lower magnesium concentrations in both femur (*p* = 0.04) and feces (*p* = 0.09). Furthermore, *Bacillus subtilis* increased plasma estrogen concentration (*p* = 0.01) and femur TNF receptor superfamily member 11b (*OPG*) expression (*p* < 0.05) but reduced plasma IL-1 (*p* < 0.01) and TNF-α (*p* < 0.01) concentrations. These results indicate that *Bacillus subtilis* could be used as a health promotor to reduce overproduction-induced inflammation and associated bone damage and to increase marketable egg production. The data provide evidence for developing a management strategy to use *Bacillus subtilis* as a feed additive to improve marketable egg production and health and welfare status of laying hens.

## 1. Introduction

Osteoporosis is one of the major threats to the health and welfare of laying hens, which causes chronic pain, bone fractures, paralysis, and mortality, resulting in poor egg production and quality as well as significant economic loss [1,2,3]. Before sexual maturity, the level of calcium (Ca) in the feed of pullets is low, which is requested only for maintaining their normal life activities, while the nutrient levels such as amino acids are high for growth of their body weight to reserve sufficient energy for laying eggs later [4]. Therefore, during the pullet period, their femur and tibia are in a hollow state. After sexual maturity begins (2 weeks before laying) along with increased estrogen levels, the skeletal state starts to change, a large amount of Ca is deposited in the bones, leading to the formation of medulla bone which services as Ca sources for laying eggs [5,6]. Thus, in order to prepare pullets to start and continue laying eggs smoothly, the composition of the feed is changed, including increasing the Ca amount from 2.2% to approximately 3.5–3.8% [7]. In addition, producers also use large particles of bone meal, limestone, or oyster shells as Ca sources [8] with phytase [9] for improvement of Ca bioavailability in laying hens by prolonging retention time in small intestines and effective quantity of Ca [10]. However, when hens become aged (40-week-old and beyond), reaching peak egg production, along with the need for a great amount of Ca for eggshell formation, which causes more Ca to be mobilized from the skeleton to the eggshell gland to remedy limited and inadequate Ca absorbed in the gut [7]. Approximately 60–70% Ca of eggshells is from diet and the rest is from bones [3]. Generally, bone loss may possibly be corrected by restoring Ca balance [11]. However, the bone-remodeling process of laying hens is affected by multiple factors, such as genetic background, individual physical and physiological characteristics, rearing environment, and nutrient status. Any alterations of either the internal, external, or both, factors will disrupt the balance between bone formation and bone resorption, mostly increasing bone resorption for eggshell formation, eventually leading to osteoporosis [5]. Thus, improvement of absorption efficiency of Ca in the intestines plays an important role in preventing osteoporosis. Probiotics function to regulate intestinal microbiota [12,13], protect intestinal integrity [14,15], and strengthen immune response [16,17]. Several probiotics have been used as alternatives to antibiotics in farm animal production [18,19]. Especially, *Bacillus subtilis* has been widely used due to its heat resistant property during processing, high survivability in acidic condition, ability to form biofilm in the small intestines, as well as due to its high stability property during storage and administration procedure.

Several studies have reported that *Bacillus subtilis* alone or combined with other bacteria can be used as growth promotors or immunomodulatory factors to increase energy retention, weight gain, and feed conversion in broiler chickens [20,21]. One of the mechanisms of its effects is to enhance feed digestion and nutrient resorption in the gastrointestinal tract (GIT) [22,23], leading to positive effects on preventing bone mass loss [24] by improving bone density and mineral content such as Ca and phosphorus (P) [24,25,26]. Compared to broilers, laying hens have a longer lifespan with a phase for great egg production, approximately 300 eggs/year or more, therefore, laying hens demand more Ca resulting in depleting structural bone over the course of production, consequently increasing risk of osteoporosis [27]. Some studies have shown that *Bacillus subtilis* promotes the absorption ability of laying hens by protecting the intestinal barrier and positively regulating intestinal microbiota composition [28,29], and increases egg quality [30,31] and mineral retention [32]. However, few studies have focused on the effects of *Bacillus subtilis* on bone pathophysiological alterations in aged laying hens and the underlying mechanisms. It is not clear if *Bacillus subtilis* can protect bone quality in laying hens with the similar effects found in broilers.

The aims of this study were to detect the influence of *Bacillus subtilis* additive in production performance, bone traits, and bone remodeling related hematology indexes in laying hens. The outcomes will provide information for better understanding the mechanisms of *Bacillus subtilis* in bone protection and the strategy using *Bacillus subtilis* to improve health and welfare of laying hens during production.

## 2. Materials and Methods

### 2.1. Animals, Housing, and Diets

All procedures in this experiment were approved by the Animal Ethics Committee of the Southwest University, Chongqing, China (permission number: IACUC-2019022519). Twenty-four 48-week-old Lohmann Pink-shell laying hens (Zhongwan Poultry Farm, Chongqing, China) were fed a regular layer diet and reared in conventional single-bird cages (one bird one cage; cage size: 40 cm × 35 cm × 35 cm) prior to the study. During the study, the birds were randomly divided into two groups (*n* = 12 replicates per treatment): a basal diet (Control, Table 1) and the basic diet supplemented with a commercial *Bacillus subtilis* (Fubon Inc., Wuhan, China) at 0.5 g/kg of feed (company recommended dose) for a 60-day trial. During the experimental period, 16 h light was provided daily, and water and feed were offered ad libitum.

### 2.2. Sample Collection

Each bird was weighed on day 0 and day 61 of the experiment. For egg production, eggs were collected in the morning (9:00–9:15) daily. The number of eggs and broken and soft-shelled eggs were recorded. The weekly total egg production and the percent of marketable eggs (normal eggs without cracks, misshapen and soft shells) [33] were calculated according to the formula: weekly egg production = total number of eggs during the week /12 birds; and the percent of marketable eggs = normal eggs/total number of eggs [34].

For egg and eggshell quality, one qualified egg was randomly collected from each cage on day 0 (the day immediately before the experiment) for measuring the initial egg and eggshell quality; and one marketable egg was per cage during day 59–60 for measuring treatment effects on egg and eggshell quality (*n* = 12 replicates per treatment).

The fresh feces samples were collected from each bird by putting a plastic bag on the bottom of each cage (*n* = 12 per treatment), at 9:00 during day 29–30 and day 59–60, respectively, to test if *Bacillus subtilis* has successfully colonized in the intestines. Feed intake was record and calculated on days 59–60 by following the procedure published previously [35].

For blood samples, 10 mL blood per bird was collected via the brachia vein into a common blood collection tube for getting serum at day 30 (*n* = 12 per treatment). After standing overnight at 4 °C, the blood was centrifuged at 3000 rpm for 15 min to obtain the supernatant. The collected serum was stored at −20 °C until biochemical indicator analyses. At day 61 (on completion of the experiment), the birds were anesthetized using 30 mg/kg of pentobarbital sodium prior to the sampling, then blood samples were collected through cardiac puncture into plasma separator tubes with EDTA and common blood collection tubes (*n* = 12 per treatment) for plasma hormones, inflammatory factors, and intestinal barrier factors and serum biochemical indicator analyses. To get plasma, the blood samples were centrifuged at 3000 rpm for 15 min, then kept at −20 °C until analysis, while the procedure for serum collection was the same as described above. For bone parameter analyses, following blood collection, the birds were euthanized immediately by cervical dislocation, and femur and tibia were collected from each bird (*n* = 12 per treatment). The right femur and tibia from each bird were frozen at −20 °C for bone strength analysis; the cancellous bone from each left distal femur was stored at −80 °C for real-time PCR analysis; and the sampled right femur, feces, and eggshell were stored at −20 °C for mineral (Ca, P, and Mg) content analysis.

### 2.3. Colonization Efficiency of Bacillus subtilis

Furthermore, 0.2 g per feces sample (*n* = 12 per group) were suspended and homogenized with 20 mL saline solution (0.9% NaCl) and then incubated at 75 °C in a water bath for 20 min. After vibration (Vortex mixer, XH-C, WoXin, Wuxi, China), each sample was diluted 10 to 1000 fold using 0.9% NaCl. A 100 μL per dilution was plated on nutrient agar media, then incubated at 37 °C for 36 h. The counts of *Bacillus subtilis* in the sample (CFU) = dilution ratio × 100 × 10 × colony number [36].

### 2.4. Egg Quality

Egg and eggshell qualities were assessed on the day before starting the experiment and day 59–60 (the last two days of the experiment) (*n* = 12 per group per time point). Egg weight, height of albumen, yolk color, and Haugh unit were measured by using an Egg Analyzer (EA-01, ORKA Food Technology Ltd., Herzliya, Israel). Eggshell strength was measured by using an Egg Force Reader (EFR-01, ORKA Food Technology Ltd., Herzliya, Israel). Eggshell thickness was tested by using an Eggshell Thickness gauge (TI-PVX, ORKA Technology, Herzliya, Israel). Egg shape index was calculated by measuring the major axis and minor axis of each egg with an Egg shape index tester (NFN383, Fujihira Industry Co., Ltd., Tokyo, Japan).

### 2.5. Hematological Measurements

Serum alkaline phosphatase (ALP), Ca, and P were analyzed using an automatic biochemistry analyzer (Olympus AU400, Tokyo, Japan), and tartrate resistant acid phosphatase (TRAP) was analyzed via a Tartrate Resistant Acid Phosphatase Assay Kit (Shanghai Biyuntian Biotechnology Co., Ltd., Shanghai, China) using a microplate reader (Olympus AU400, Tokyo, Japan). Additionally, the concentrations of interleukin (IL)-1, IL-6, tumor necrosis factor-alpha (TNF-α), 1.25-dihydroxy vitamin D (1.25-(OH)_2_D_3_), thyroid hormones (PTH), calcitonin (CT), lipopolysaccharide (LPS), D-lactic acid (D-LA), and estrogen (E2) were detected by using relative ELISA kits (Xiamen Huijia Biotechnology Co., Ltd., Fujian, China).

### 2.6. Bone Traits Analysis

The muscle of both right femur and tibia were removed, and then the bones were weighed and scanned for bone mass and densitometry by using an electronic analytical balance (JA2003A, JingTian, Shanghai, China) and the InAlyzer (MEDIKORS, Seongnam, Korea) (*n* = 12 per group). Relative mass of each bone = bone weight/body weight. Breaking strength of femur and tibia was measured using the Universal Test Machines (LR10K Plus, Lloyd Instruments Ltd., Wokingham, UK) with adopted three points bending method and applied mean extension rate of 10 mm/min until failure, and the maximum load (the point on the load deformation curve, beyond which plastic damage of the bone will be occurred), maximum stress (the internal resistance of bone to external forces, beyond which bone fractures will be occurred), maximum strain (the ratio of change in the length to the original size of bone under maximum stress), stiffness (the result is calculated as being the gradient of the modulus line on a load vs. extension graph), and Young’s modulus (the ratio of the stress to strain in the linear range of the curve) were calculated [35,37,38].

### 2.7. Real-Time PCR

Total RNA was randomly extracted from 6 of 12 sampled cancellous bones of the left distal femur (*n* = 6 per group) using the Bone tissue RNA Extract Kit (Coolaber technology co., Ltd., Beijing, China). The quality and concentration were examined using NanoPhotometer (P330, Implen, Munich, Germany). cDNA was synthesized from 1 μg of total RNA using NovoScript Plus All-in-one 1st Strand cDNA Synthesis SuperMix (gDNA Purge) (Novoprotein Scientific Inc., Suzhou, China). The mRNA levels of TNF receptor superfamily member 11b (*OPG*), collagen type I alpha 2 chain (*COL1A2*), sclerostin (*SOST*), TNF superfamily member 11 (*RANKL*), TNF receptor superfamily member 11a (*RANK*), and glyceraldehyde-3-phosphate dehydrogenase (*GAPDH*) were measured by using real-time PCR. The *GAPDH* was used as the reference gene. The expression of each target gene was determined by using the 2^−ΔCt^ method. The specific formula was: ΔCt = Ct_target gene_ − Ct_housekeeping gene_. The sequences of primers were synthesized by Invitrogen Biotechnology (Shanghai, China) and listed in Table 2.

### 2.8. Ca and P Content Analysis

The sampled eggshells, feces, and femur were dried in an air oven at 105 °C until constant weight was reached. The weighed 0.1 g sample was placed into a crucible and carbonized on electric heating plates at 200 °C. Each sample was then ashed in a muffle furnace (FO811C, Yamato scientific America, Inc., Tokyo, Japan) at 550 °C for 3 h, and the ash was weighed to gain the inorganic mass. The ash was dissolved with nitric acid and diluted up to 50 mL with ultra-pure water. The contents of Ca, P, and Mg were analyzed using an inductively coupled plasma-optical emission spectrometer (ICP-OES) (iCAP 7000 SERIES, Thermo, Waltham, MA, USA) [39,40,41].

### 2.9. Statistical Analysis

All data were analyzed by using the Statistical Package for the Social Science (SPSS). The overall differences between the two groups were analyzed through independent-sample *t*-test and multiple comparisons of means produced by LSD. The effects of *Bacillus subtilis* and age on the biochemical indexes of bone metabolism were analyzed by two-way ANOVA. When there was an interaction effect, the *t*-test was used to compare the differences between groups. The correlation analysis between bone traits and Ca or P in femur and serum was performed using the Spearman procedure via the two-tailed test. *p* ≤ 0.05 were considered statistically significant and 0.05 < *p* < 0.1 was considered a trend difference.

## 3. Results

### 3.1. Colonization Efficiency of Bacillus subtilis

The results showed that dietary *Bacillus subtilis* supplement significantly increased the count of colonized *Bacillus subtilis* in the feces of treated group compared to the control group during the experiment period (*p* < 0.01; Table 3).

### 3.2. Egg Production and Egg Quality Parameters

There were no treatment effects on egg production (*p* > 0.05). In addition, there was no difference in egg quality between the control and treated groups before the experiment (data not shown). However, *Bacillus subtilis* feeding significantly increased the number of marketable eggs during the entire experiment (*p* < 0.01, Table 4) by reducing broken (17.08% vs. 7.12%, control vs. treated group, *p* < 0.01) and soft-shelled eggs (0.69% vs. 0%, control vs. treated group, *p* < 0.05), while there were no treatment effects on the egg weight, height of albumen, yolk color, Haugh unit, eggshell thickness, and strength as well as eggs’ shape index (*p* > 0.05).

### 3.3. Hematological Analysis

*Bacillus subtilis* did not affect serum Ca concentration, ALP, and TRAP activities (*p* > 0.05; Figure 1A,C,D) but reduced P concentration (*p* < 0.01, Figure 1B) in laying hens at day 61. The serum activity of ALP (*p* = 0.06) had a trend of decrease and TRAP activity (*p* < 0.01) significantly increased with age in laying hens (Figure 1C,D). Meanwhile, at day 61, there was a trend of lower plasma CT in *Bacillus subtilis* fed birds (*p* = 0.07; Figure 2A). In addition, *Bacillus subtilis* increased plasma concentrations of E2 (*p* = 0.01) without effects on plasma concentrations of IL-6, 1,25-(OH)_2_D_3_, PTH, D-LA, and LPS (*p* > 0.05; Figure 2A–C), but inhibited plasma IL-1 (*p* < 0.01; Figure 2B) and TNF-α (*p* < 0.01) concentrations.

### 3.4. Bone Pathophysiological Parameters

Compared with the control group, *Bacillus subtilis* administration did not change femur absolute and relative mass and density (*p* > 0.05; Table 5) but improved femur maximum stress (*p* = 0.01), stiffness (*p* < 0.01), and Young’s modulus (*p* < 0.01) in birds. Moreover, there was an upward trend of femur maximum load and a downward trend of femur maximum strain, respectively, in *Bacillus subtilis* fed birds (*p* = 0.06). However, *Bacillus subtilis* supplement did not affect other measured biomechanical properties of tibia, except for a slight increase in tibial density (*p* = 0.07). Furthermore, *Bacillus subtilis* supplement improved the expression of *OPG* mRNA (*p* = 0.02; Figure 3), leading to a greater *OPG*/*RANKL* ratio (*p* = 0.04), while there were no treatment effects on the expression of *RANKL*, *RANK*, *SOST,* and *COL1A2* mRNA.

### 3.5. Ca and P Content Analysis

Compared with the control group, bone P content was significantly increased (*p* < 0.05; Table 6) in the *Bacillus subtilis* fed group. The *Bacillus subtilis* fed group also had a lower concentration of bone Mg (*p* = 0.04) and a trend of higher fecal Mg (*p* = 0.09). However, there were no treatment effects on the contents of fecal Ca and P, bone Ca and ash weight, and eggshell P (*p* > 0.05).

### 3.6. Correlation between Ca or P and Bone Quality

The femoral P but not Ca level was positively correlated with femur maximum stress (*r* = 0.55, *p* = 0.02; Table 7), stiffness (*r* = 0.45, *p* = 0.05), Young’s modulus (*r* = 0.56, *p* = 0.014), and femoral Ca (*r* = 0.80, *p* < 0.01) as well as being negatively correlated with serum P (*r* = −0.59, *p* < 0.01), serum Ca level was unrelated to any bone quality indexes. However, serum P level was negatively correlated with stiffness (*r* = −0.71, *p* = 0.001) and Young’s modulus (*r* = −0.58, *p* = 0.009).

## 4. Discussion

With age (beyond 40 weeks of age), especially into the later stage of egg production, their intestinal absorption and utilization of minerals including Ca are gradually declined [42,43], which could be due to the decrease of relative hormones and their receptors [39,44] or the change of gut epithelial architecture reducing absorption surface [45]. To maintain the need of Ca for egg production, layers may mobilize extra Ca stored in medullary bone. Approximately 25–40% of the eggshell Ca is from skeletal stores [46]. This process causes negative balance between bone formation (osteoblasts) and resorption (osteoclasts) with the enhanced osteoclast activity and releasing of large amounts of TRAP [47,48], resulting in significant expending of medullary bone at the expense of structural bone (cancellous and cortical bone), eventually leading to osteoporosis. To adapt to Ca shortage, laying hens adjust Ca usage for eggshell formation [49,50] by either decreasing eggshell thickness and strength [51], reducing egg production, or in combination [49].

As the egg cycle extended, the numbers of soft-shelled and cracked eggs increased, which may be attributed to reduction of Ca absorption, stress, or poor physical quality [52,53,54]. The current results indicate that hens fed *Bacillus subtilis* had a lower incidence of both soft-shelled and cracked eggs but there was no strong evidence of the improvement of eggshell strength and thickness [55]. Inconsistent findings about effects of *Bacillus subtilis* on egg quality have been reported. Some studies indicated that *Bacillus subtilis* could improve albumen height and Haugh unit [31,56]. On the contrary, our findings as well as some of others showed that egg quality was not affected by *Bacillus subtilis* supplement [57,58]. Previous study reported that probiotic efficiency as a growth promotor is affected by multiple factors, such as the type of probiotics, the length of feeding time and concentration [31], the isolation sources of probiotics, breed of laying hens, and their life-stage [59]. In addition, Alfonso-Carrillo et al. [54] indicated that the promotion of egg production, eggshell quality, and bone traits in laying hens can occur independently through dietary treatment. Therefore, we speculate that *Bacillus subtilis* is more likely to improve bone traits rather than egg or eggshell quality.

Stiffness and Young’s modulus have been used as indicators indicating the capability of bone to resist deformation, which is mainly determined by cortical bone density. Increased bone mass can improve bone strength and stiffness [60]. In the present study, dietary *Bacillus subtilis* supplement significantly improved bone strength, stiffness, and Young’s modulus of femurs, and while *B**acillus subtilis* changed both bone stress and strain in laying hens. Previous study has demonstrated that the fluctuations caused by changed bone strain and stress are mainly due to increased inhomogeneous mineral distribution in trabecular bone, namely, a higher bone stress but lower strain [61]. The outcomes of the present research suggest that *Bacillus subtilis* is able to prevent bone mass loss and reduce fracture risk. In addition, Ca and P are two necessary minerals for bone mineralization and bone homeostasis. Phosphorus is mainly deposited in bone in the form of hydroxyapatite with Ca and is involved in regulation of bone cells’ functions and ALP metabolism [62,63]. It has been revealed that besides being absorbed from diet, the skeletal system is a main source of minerals for eggshell formation [64]. It means that P is also mobilized from bone to blood for egg production. Shao [65] reported that the plasma concentration of P is an indicator of bone mineralization, and even a slight decline shows that bone mineralization is in a positive state with improved bone strength. This is consistent with the current results that the *Bacillus subtilis* reduced serum P concentration while it improved bone P content, and bone P content had a great correlation with serum P, and was positively associated with bone Ca content, stiffness, and other physical parameters. These results show that *Bacillus subtilis* provide the beneficial effects on bone health via regulation of P metabolism.

Estrogen is a kind of bone regulating hormone. Enterohepatic circulation is the main way to regulate E2 metabolism due to its bioconversion in the liver, conjugated E2 is mostly excreted with bile, but some is hydrolyzed by gut microbiota, and reabsorbed in the intestines for various biological processes including bone remodeling [66,67]. Similar to the current results, Zhou [68] reported that adding *Bacillus amyloliquefaciens* increased serum E2 level in laying hens. We speculate that the increase of E2 level is attributed to the colonization and functional activities of *Bacillus subtilis* in the intestines. Furthermore, E2 has been recognized as functionally preventing osteoporosis in postmenopausal women [69,70] and old laying hens [44]. Previous results have shown that E2 could decrease synthesis of inflammatory cytokines and regulate the OPG/RANKL/RANK system to prevent bone loss [71]. Estrogen deficiency leads to high productions of TNF and RANKL in bone marrow and small intestines [72,73]. However, probiotics supplements can protect ovariectomized mice from bone lose [74,75]. These evidences suggest the possible protective effects of probiotics and E2 on bone. Consistently, *Bacillus subtilis* supplement significantly reduced hens’ plasma pro-inflammatory factors, IL-1 and TNF-α, and increased the *OPG* gene expression in the present study.

The IL-1 and TNF-α are mainly secreted by peripheral monocytes, macrophagocytes, and T lymphocytes [76]. Reduced serum IL-1 and TNF-α in the hens suggests that *Bacillus subtilis* prevents the inflammatory process induced by aging-related immune response. Similarly, Lee et al. [77] found that probiotics prevent the progression of inflammation-associated intestine injury through intervening in the expression of inflammatory molecules, such as IL-1 and TNF-α, to maintain intestinal health. These pro-inflammatory factors play important roles in activation of osteoclasts and related bone resorption [78]. Moreover, the OPG/RANKL/RANK system has been identified as the dominant and final mediator of osteoclastogenesis [79]. OPG, as the only known decoy receptor of RANKL, is located on osteoblasts and can be upregulated by E2 [80]. OPG competitively binds with RANKL [78] to prevent osteoclasts from being activated by the signals released from the RANKL/RANK pathway, consequently, it inhibits bone resorption and bone loss [81]. The ratio of OPG/RANKL has been recognized as an anti-osteoclast indicator [82,83]. Taken together, these results indicate that the immune system is involved in regulation of bone remodeling (formation and resorption), especially influencing the activation of the OPG/RANKL/RANK pathway [71,84], for example, IL-1 indirectly inhibits *OPG* mRNA expression [85] without any impact on RANKL expression, and stimulates osteoclastogenesis resulting in bone loss [86]. Therefore, it is possible that *Bacillus subtilis* protects hen bone health through improving the E2 metabolism, inhibiting synthesis of pro-inflammatory factors, and increasing OPG synthesis.

## 5. Conclusions

Low-grade systemic inflammation is one of the major health and production issues in laying hens, resulting in osteoporosis with chronical pain, bone fractures, and poor egg production with significant economic loss. This study showed that *Bacillus subtilis* as an additive to laying hens’ diets can increase marketable egg production and bone traits by increasing P usage, improving estrogen metabolism, inhibiting pro-inflammatory factor synthesis, and increasing *OPG* expression. Thus, *Bacillus subtilis* can be a valuable feed supplement to improve the welfare and prevent osteoporosis in laying hens.

## Figures and Tables

**Figure 1 animals-11-02041-f001:**
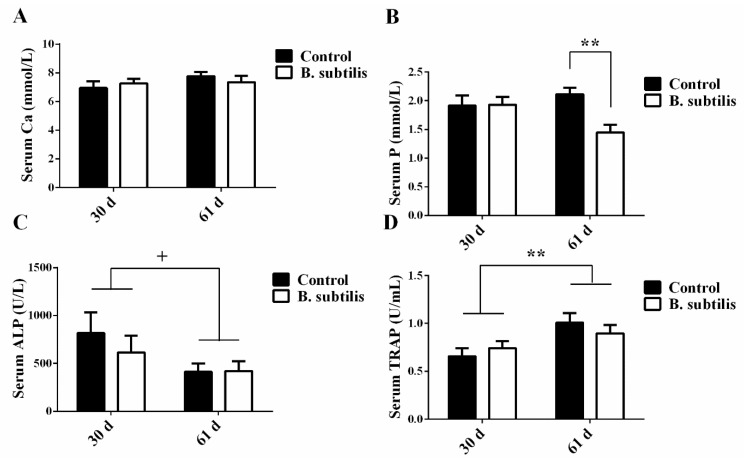
The effects of *Bacillus subtilis* and age on biochemical indicators of bone metabolism in laying hens. (**A**) Ca concentration in serum at day 30 and day 61. (**B**) P concentration in serum at day 30 and day 61. (**C**) ALP concentration in serum at day 30 and day 61. (**D**) TRAP concentration in serum at day 30 and day 61. Data are presented as mean ± SEM (*n* = 12 per group) ** means *p* ≤ 0.01; and ^+^ means 0.05 < *p* < 0.1. ALP: alkaline phosphatase; Ca: calcium; P: phosphorus; TRAP: tartrate-resistant acid phosphatase.

**Figure 2 animals-11-02041-f002:**
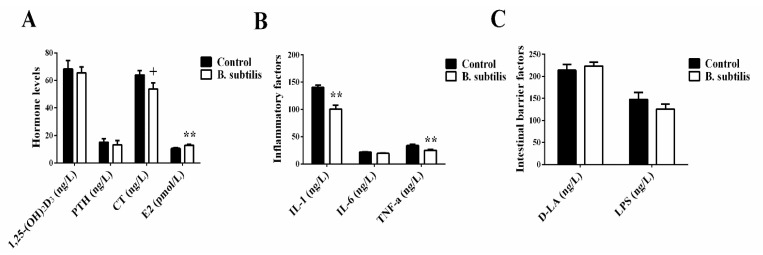
The effects of *Bacillus subtilis* on inflammatory factors, hormone levels, and intestinal barrier factors in laying hens at day 61. (**A**) Hormone levels in plasma. (**B**) Inflammatory factors concentrations in plasma. (**C**) Intestinal barrier factors concentrations in plasma. Data are presented as mean ± SEM (*n* = 12 per group). ** means *p* ≤ 0.01; and ^+^ means 0.05 < *p* < 0.1. PTH: thyroid hormones; CT: calcitonin; E2: estrogen; IL-1: interleukin-1; IL-6: interleukin-6; TNF-α: tumor necrosis factor-alpha; D-LA: D-lactic acid; LPS: lipopolysaccharide.

**Figure 3 animals-11-02041-f003:**
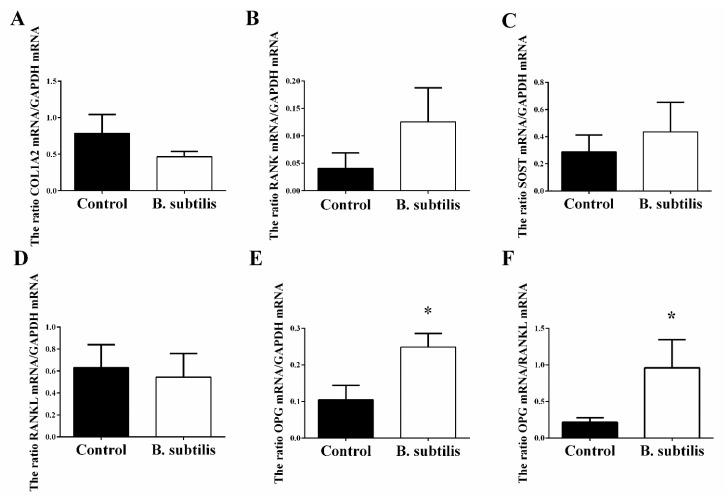
The effects of *Bacillus subtilis* on bone mRNA expressions in laying hens. (**A**) The change of *COL1A2* mRNA expression. (**B**) The change of *RANK* mRNA expression. (**C**) The change of *SOST* mRNA expression. (**D**) The change of *RANKL* mRNA expression. (**E**) The change of *OPG* mRNA expression. (**F**) The ratio of *OPG*/*RANKL* mRNA expression. Data are presented as mean ± SEM (*n* = 12 per group) * means *p* ≤ 0.05. *COL1A2*: collagen type I alpha 2 chain; *RANK*: TNF receptor superfamily member 11a; *SOST*: sclerostin; *RANKL*: receptor activator for nuclear factor-κ B ligand; *OPG*: TNF receptor superfamily member 11b.

**Table 1 animals-11-02041-t001:** Composition of the basal diet for layers.

Item	Factor	Control Diet	*B. subtilis* Diet
Ingredient	Corn (%)	65	65
	Soybean (%)	22	22
	Shell powder (%)	8.9	8.9
	Zeolite powder (%)	1.1	1.1
	3% premix ^1^ (%)	3	3
	*Bacillus subtilis* (g/kg)	0	0.5
Nutrient composition	Crude protein (%)	15.05	15.05
	Calcium (%)	3.7	3.7
	Energy (MJ/kg)	11.57	11.57

^1^ Premix contains multivitamins, complex trace elements, DL-methionine, calcium hydrophosphate, light calcium carbonate, sodium chloride, phytase, choline chloride, antioxidant, and zeolite powder. Ingredients per kilogram of premix: vitamin A, 220,000–330,000 IU; vitamin D_3_, 55,000–85,000 IU; vitamin E: ≥320 mg; vitamin K_3_, 40–140 mg; vitamin B_1_, ≥75 mg; vitamin B_2_, ≥155 mg; vitamin B_6_, ≥75 mg; thiamine nitrate, ≥80 mg; calcium pantothenate, ≥155 mg; nicotinamide, ≥850 mg; iodine, 5–15; iron 2000–6000 mg; zinc, 2400–4830 mg; manganese, 2930–4820 mg; copper, 267–667 mg; selenium, 5–15 mg; calcium, ≥8%; total phosphorus, ≥3.3%; sodium chloride, 7–14%; methionine, ≥2.3%.

**Table 2 animals-11-02041-t002:** Real-time PCR primers and amplified PCR product size.

Gene	GenBank ID	PCR Primers Sequence (5′ to 3′)	PCR Products (bp)
*OPG*	NM_001033641.1	F: GTTCCTACTCGTTCCACACC R: GCTCTTGTGAACTGTGCCTTTG	115
*RANKL*	XM_015275777.2	F: CTGGAACTCGCAAAGTGAACCT R: TTTCCCATCACTGAACGTCATATTT	86
*SOST*	XM_025144077.1	F: TTGTCTGTATTCGTCTCGCTAT R: AACGTCCTTTCTGAGTCACCT	180
*COL1A2*	NM_001079714.2	F: GGCTTTGATGCAGAATACTACCG R: GTTGTTCAATGTTTTCAGAGTGGC	90
*RANK*	XM_004939689.3	F: GCCATGTCCCAGAGGATACT	87
		R: GCCAATCCCAGAGCTGAACA	
*GAPDH*	NM_204305.1	F: TTGACGTGCAGCAGGAACAC R: ATGGCCACCACTTGGACTTT	124

*OPG*: TNF receptor superfamily member 11b; *RANKL*: receptor activator for nuclear factor-κ B ligand; *SOST*: sclerostin; *COL1A2*: collagen type I alpha 2 chain; *RANK*: TNF receptor superfamily member 11a; *GAPDH*: glyceraldehyde-3-phosphate dehydrogenase.

**Table 3 animals-11-02041-t003:** The capability of colonization of *Bacillus subtilis* expressed in feces.

Time	Lg (Control)	Lg (*B. subtilis*)	SEM	*p*-Value
D 30	2.67	6.41 **	0.98	0.007
D 60	3.31	6.08 **	0.75	0.007

Values are represented by mean ± SEM (*n* = 12 per group), ** means *p* ≤ 0.01. Lg: the original count of colonized *Bacillus subtilis* data were presented by lg conversion.

**Table 4 animals-11-02041-t004:** Effects of *Bacillus subtilis* on egg production and egg qualities of laying hens from 48 to 57 weeks of age.

Parameters	Control	*B. subtilis*	SEM	*p*-Value
48 week body weight (kg)	1.69	1.70	0.08	0.32
57 week body weight (kg)	1.72	1.73	0.06	0.76
Feed intake (g)	120.33	119.28	10.95	0.93
Egg production, %	80.64	78.26	1.71	0.18
Marketable eggs, %	76.19	88.74 **	2.29	<0.01
Egg weight (g)	62.08	62.8	1.80	0.69
Egg shape index	1.35	1.34	0.02	0.64
Height of albumen (mm)	6.55	6.48	0.34	0.84
Yolk color	5.08	4.58	0.35	0.16
Haugh unit	79.75	79.05	2.46	0.78
Eggshell thickness (mm)	0.47	0.48	0.00	0.45
Eggshell strength (kg)	3.54	3.47	0.41	0.87

Data are presented as Mean ± SEM, ** means *p* ≤ 0.01 (*n* = 12 per group).

**Table 5 animals-11-02041-t005:** Effects of *Bacillus subtilis* on the parameters of bone strength.

Bone Strength Parameters	Control	*B. subtilis*	SEM	*p*-Value
-	Femur		-	
Absolute mass (g)	8.67	8.47	0.32	0.54
Relative mass%	50.63	49.31	1.85	0.48
Maximum load (N)	118.85	143.5 ^+^	12.1	0.06
Maximum stress (MPa)	115.63	211.90 **	32.4	0.01
Maximum strain	0.014	0.012 ^+^	<0.0001	0.06
Stiffness (N/m)	148,607	205,039 **	13,906	<0.01
Young’s modulus (MPa)	10,226.7	28,910.1 **	3586.1	<0.01
Density (g/cm^2^)	0.49	0.43	0.11	0.57
-	Tibia		-	
Absolute mass (g)	9.93	9.52	0.35	0.25
Relative mass%	58.10	55.05	1.91	0.13
Maximum load (N)	97.86	101.34	8.98	0.70
Maximum stress (MPa)	191.31	202.39	35.93	0.76
Maximum strain	0.040	0.037	0.001	0.31
Stiffness (N/m)	44,657.21	46,069.66	2705.6	0.61
Young’s modulus (MPa)	6372.69	6679.99	1299.8	0.82
Density (g/cm^2^)	0.28	0.38 ^+^	0.05	0.07

Data are presented as mean ± SEM. (*n* = 12 per group), ^+^ means 0.05 < *p* < 0.1; and ** means *p* ≤ 0.01. SEM: Standard Error of Mean.

**Table 6 animals-11-02041-t006:** The effects of *Bacillus subtilis* on the Ca, P, and Mg content in laying hens.

Sample	Parameters	Control	*B. subtilis*	SEM	*p*-Value
Eggshell	Ca (mg/g)	281.77	270.56	6.91	0.12
-	P (mg/g)	1.04	1.02	0.19	0.94
-	Mg (mg/g)	3.15	3.04	0.12	0.49
Excretion	Ca (mg/g)	53.76	75.54	12.46	0.10
-	P (mg/g)	13.26	12.81	1.64	0.79
-	Mg (mg/g)	3.51	4.26 ^+^	0.42	0.09
Bone	Ca (mg/g)	240.98	243.12	8.43	0.80
-	P (mg/g)	94.60	103.78 **	3.17	<0.01
-	Mg (mg/g)	3.39	3.09 *	0.14	0.04
-	bone ash (%)	58.41	59.87	3.20	0.65

Data are presented as mean ± SEM (*n* = 12 per group), ^+^ means 0.05 < *p* < 0.1; * means *p* ≤ 0.05; and ** means *p* ≤ 0.01.

**Table 7 animals-11-02041-t007:** The correlation between bone P and Ca contents as well as serum P and Ca levels and bone physical parameters in laying hens.

Femur	Correlation Analysis	Serum Ca	Serum P	Femoral Ca	Femoral P
Maximum load	r	0.26	−0.32	0.21	0.42
	*P*	0.28	0.18	0.40	0.07
Maximum stress	r	0.34	−0.44	0.25	0.55 *
	*P*	0.16	0.06	0.30	0.02
Maximum strain	r	0.13	0.28	−0.08	−0.26
	*P*	0.59	0.24	0.75	0.28
Stiffness	r	−0.07	−0.71 **	0.11	0.45 *
	*P*	0.77	<0.01	0.67	0.05
Young’s modulus	r	0.16	−0.58 **	0.12	0.56 *
	*P*	0.51	<0.01	0.63	0.01
Serum Ca	r	-	0.35	−0.13	−0.11
	*P*	-	0.15	0.59	0.67
Serum P	r	-	-	−0.31	−0.59 **
	*P*	-	-	0.20	<0.01
Femoral Ca	r	-	-	-	0.80 **
	*P*	-	-	-	<0.01

Values are represented as the correlation coefficients between femur qualities and Ca or P in femur and serum, respectively (*n* = 12 per group), * means *p* ≤ 0.05; and ** means *p* ≤ 0.01.

## Data Availability

Not applicable.

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
