# Peer review of "Effects of Bacillus subtilis on Production Performance, Bone Physiological Property, and Hematology Indexes in Laying Hens"

_animals, 2021, doi:10.3390/ani11072041_

Round 1

Reviewer 1 Report

Some important observations that should be adjusted in the introduction:

Introduction:

  1. The authors must cite the importance of pullet rearing and nutrition
  2. The importance of calcium source including mineral availability and feed particle are essential for this scenario
  3. Absorption rates can be due to the individual characteristics
  4. The efficiency of probiotics in laying hens are different from broilers. PLease, focus your literature review in recent laying hens’ citations.

Materials and Methods

  • 24 hens seems to be a low group to attend the purpose of the experiment
  • White Plymouth laying hens, did you use a original hen or a genetic line ?
  • In my opinion, 60 days for this experiment are low period for this kind of evaluation
  • Were each of the birds considered as one replicate ?
  • Please define marketable eggs
  • Why didn’t you supplement Choline in the diets ?
  • 24 eggs were collected for quality evaluation ?

Results

  • Egg production and Egg Quality Parameters were not different, result of a low replicate numbers
  • Figure 1 and 2 – did you compare Only in the ages or the age effect (61 x 61 days or 30 x 61 days) ? it is important to verifiy this effect
  • I’ve tried to understand all the variables expressed in Table 5 and most of them are not defined in materials and methods: like Maximum load, Maximum stress, Maximum strain Stiffness, Young's modulus, and others
  • In table 6, to understand the excreted mineral content, it should be controlled the minearl intake to estimate the absorption rate

Discussion

  • For the phrase “With hens become old (beyond 40 weeks of age),...” please, correct English writing
  • From line 338 to 353, the authors discuss the results based in microbiota composition, but this analysis was not done. Then, I recommend to include this data or change the phrases

Conclusion

  • The authors start the conclusion with the phrase: “Osteoporosis in laying hens due to great egg production and related imbalance 375 mineral metabolism”, but the production in this experimente was not high and metabolism was not completely discussed.
  • As the authors did not include a digestibility trial with biochemical blood analysis, it is just na inference, that is not adequated.

Reviewer 2 Report

The paper is presented very well, easy to follow and provides input to the scientific world. However, there are few grammatical corrections.

  1. Line 59: correct and or as "and/or"
  2. Line 66: correct the line as "Especially, Bacillus subtilis has been widely used due to its heat resistant property during processing, high survival in acidic condition, ability to form biofilm in the small intestines as well as due to its high stability property during storage and administration procedure".
  3.   Line 111: correct experience as "experiment"
  4. Line 116: remove 'A' before 10 mL
  5. Line 135: remove a before 75oC
  6. Line 136: correct vibrated as "vibration"
  7. Line 141: remove word immediately
  8. Line 189: add "was" after 0.1 g sample
  9. Line 191: correct weighted the ash as "the ash was weighted to gain"
  10. In Table 4: Could you please write the age of the bird in parameter initial body weight and body weight to make it clear.
  11. Line 222: n=12 per grou correct it to "n=12 per group"
  12. Line 242: what does the sign "+" in +0.05<P<0.1 stands for
  13. In Table 5: There is still "+" sign in B. subtilis column for maximum strain and density, could you please clarify it in the section under the table
  14. Line 278: Correct has with "was"
  15. Line 285: correct With hens become old with "With age (beyond 40 weeks of age)"
  16. In line 289: replace requested Ca with "need of Ca"
  17. In line 289: correct sentence layers may extra mobilize to "layers may mobilize extra"
  18. Line 325: replace forming eggshell with "eggshell formation"
  19. Line 332: replace evidence with "shows"
  20. Line 332: replace proves the with "provide"
  21. Line 335: correct sentence due to its bioconversion occurs in the liver as "due to its bioconversion in liver"
  22. In line 375: replace great with "higher"

Reviewer 3 Report

The manuscript is suitable for publication after correcting the footer of the ration table.

Reviewer 4 Report

Comments to the Authors of manuscript number: animals-1247986 entitled “Effects of Bacillus subtilis on production performance, bone physiological property, and hematology indexes in laying hens”.

Authors present the study performed on the old laying hens supplemented with Bacillus subtilis.

  1. Authors present the problem in bone health in laying hens. It is very vell presented in the introduction.
  2. They also show shortly the advantages of the bacteria use.
  3. L 105 is it the day of the study or their life?
  4. Authors should explain how many hens were in one pen.
  5. What was a sample in each parameter?
  6. L 110 – it should be explained from how many hens were these eggs collected. Were they from 12 hens? 2 from 1 hen?
  7. L 112 – the same as upper
  8. L 113- the same – it should be explained
  9. L 114- how the feces were collected? Were hens held separately?
  10. L 128 – not tibia bone but shortly tibia or in plural tibiae; femur, femora
  11. L 129 -130 – one bone from one bird?
  12. L 157 – small letter
  13. L 166 – if is possible to determine the load during elongation? How to use the three bending test for elongation?
  14. They should explain how They chose the term of the study and the dose of bacteria used. It should be clearly explained.

It is very good study. I met in my own scientific work, Reviewers who in such case as here presented, that means only one dose and only two groups, they wrote that it is not good study, and then the manuscript was rejected.

In my mind, Authors should complete lacking information, and except for the lack of the other doses, it can be published.

Round 2

Reviewer 1 Report

Theo suggestions made in the First evaluation were considered from the authors.

Iy opinion, despite of some itens pointed, it seems ok now

Author Response

Thank you very much for your recognition of the article.

Reviewer 4 Report

I have no comments

Author Response

(The authors gave the same response as above.)
